# Application of a Program to Improve Personal Development in Future Physical Education Teachers of the Degree in Education and Its Relationship with Wisdom

**Milagros Arteaga-Checa [1]**, **María Victoria Palop-Montoro [2]** and **David Manzano-Sánchez [3],***

[1]  Faculty of Humanities and Education Sciences, University of Jaén, 23071 Jaen, Spain; marteaga@ujaen.es
[2]  Department of Health Sciences, Catholic University of Murcia, 30107 Murcia, Spain; mvpalop@ucam.edu
[3]  Faculty of Sports Sciences, University of Murcia, 30100 Murcia, Spain
\*  Correspondence: david.manzano@um.es

**Abstract:** The objective of the present study was to apply an intervention program based on emotional education and self-knowledge in students of the degree in Education to verify changes in wisdom to improve their psychological health and emotional well-being. For this, Three-dimensional Wisdom Scale (3S-WS) was administered before and after the intervention, analyzing aspects related to affective, cognitive and reflective wisdom. The sample consisted of 100 students (40 men and 60 women, aged between 20 and 29 years). After the intervention program, students improved reflective wisdom without an identifiable difference between sexes. On the other hand, men had higher values in all variables than women. In conclusion, the program to improve personal development and self-awareness could be useful to improve wisdom (especially reflective wisdom) in third and fourth year students of the degree in Education specializing in Physical Education. At the same time, it is intended that these students understand the foundations of the intervention so that in the future it can be replicated in their classrooms and contribute to the sustainable development of the 2030 Agenda.

**Keywords:** cognitive domain; motivational education; physical education degree; social ambit

## 1. Introduction

Education must train students in identity and values to obtain adequate adaptation and social integration. For this purpose, we highlight in this article the importance of training and progressing in "personal development and awareness", considering it a pillar on which to support the personality maturation process, to promote a positive attitude and gain confidence and responsibility within both the academic and personal spheres [1,2].

Personal development acquires importance as a formula to develop emotional aspects in the educational field, since, as Fernández-Berrocal [3] explains, reality has changed, and it no longer has as a priority to develop intellectual and academic aspects, but rather, it is necessary to face emotional and social aspects at the educational level. Here, various investigations have shown that students lack the psychological resources that allow them to successfully cope with problems associated with negative behavior and emotional difficulties at school [4–7].

The communicative, creative and social possibilities that derive from the treatment of corporal expression justify the importance of working on this block of contents within the framework of physical education. Corporal expression is part of the current educational reality and allows the development of the expressive and communicative capacity of the person. The elements to consider for the development of expressive capacity are knowledge and personal development (disinhibition, personal expressive-segmental knowledge and personal knowledge-adaptation to the environment), interpersonal communication (verbal

and non-verbal communication) and introjective communication (information about our inner self) [8].

In the field of physical education, body expression should contribute to the integral development of the individual by enhancing the knowledge and development of body language through different techniques [9], since it is a "language that becomes an educational material and is used for the potential development of the expressive capacity of the human being, promoting personal knowledge, interpersonal communication and the externalization of the individual's internal feelings (introjective communication), to through gestures, postures and expressive movements" [10].

One of the essential values for true personal development is wisdom (Figure 1), understood as the degree of learning associated with personal and level development, implying an exceptional amount of knowledge, judgment and advice applied to important, complex and uncertain issues related to life and its meaning [2] and taking into account the limits and uncertainties in which all knowledge moves [11]. Other authors link wisdom to aspects related to personality and affections, to the achievement of psychological maturity and to attributes such as transcendence, compassion, prudence and commitment to other people [12,13].

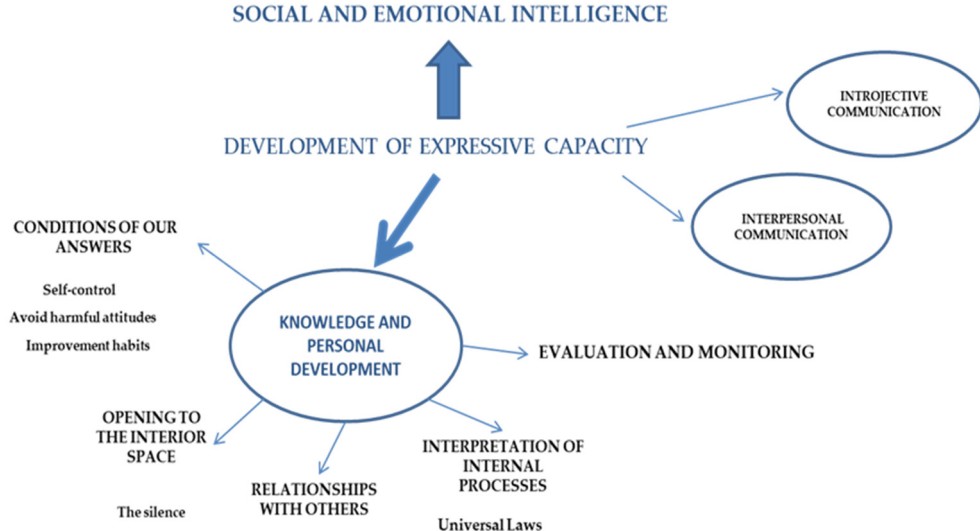

**Figure 1.** Components of wisdom (own elaboration from reference [10]).

Several authors have described the dimensions of wisdom. Webster [14] differentiates five dimensions in wisdom: experience, emotional regulation, reminiscence, openness to experience and humor. Although these are not without debate, they do seem to be essential components of wisdom, both necessary and sufficient and which constitute the predictors and consequences. A basic definition is that wisdom is composed of a cognitive aspect (general), a reflective aspect (related to oneself) and an affective aspect (related to others) [15]. Montávez's research [16] is also noteworthy, adding that the process of discovery and development of individual expressiveness is made up of three phases necessary for its integration into the reality of each person. The phases established by the author are: "Awareness of the body (physical bases), emotional experience (expressive bases) and the consolidation of expressive abilities".

Wisdom is a rare human virtue and a form of advanced cognitive and emotional development derived from experience that cannot be artificially fostered [17]. At the same time, emotional well-being is shaped by personality traits, without knowing with certainty which processes exactly influence it. Apparently, cognitive regulation serves to improve well-being, suggesting that interventions that help reduce unpleasant emotional experiences associated with anxiety and depression also promote greater happiness and emotional well-being. Emotional well-being work can help avoid negative effects such as depression [18], as well as more flexible thinking and the ability to find positive aspects

in stressful situations [19]. However, the most important thing is that there are authors who consider emotions as high-intensity affective responses that can be managed through training [20]. Thus, the review by Sánchez-Álvarez [21] shows how research on emotional well-being has increased significantly in recent decades given the great interest in improving the well-being of the person, the sample of university students being one of the most used in most investigations.

Focusing on the development of the person, Goleman [22] labeled knowledge and personal development with the name of "self-knowledge" or "emotional self-awareness". For this author, emotional self-awareness is the first component of emotional intelligence, of which he states that "knowing how one feels implies becoming aware of internal states, resources and intuitions, making an objective assessment of oneself and being able to recognize our own strengths and weaknesses".

For these skills to work, the spontaneous and natural expression of the body as a means of development takes on a special relevance, allowing a greater development in society of the person and the achievement of personal fulfillment [23,24]. In this way, one of the most important needs of the human being is to express himself, and frustration in this area can contribute to behavioral disturbances, mainly in early ages, since what happens in the first years of life will mark the evolutionary development of the child [10]. Thus, the awareness and sensitization when learning to observe and analyze our behaviors and those of others through body expression can help develop the wisdom that will help us understand life [2].

It is from childhood that we are filled with beliefs that are initially patterns that have been reinforced by adults and will configure a belief system to adapt to the world. Parts of these beliefs are based on messages of ignorance that produce perceived limitations, and it is the ego that is responsible for making us believe that we are safer within that imprisonment than outside of it [2]. To be able to dismantle the ingrained patterns that are the result of having seen life in one way for a long time, it requires training, and it takes time to dissolve them. In addition, there is considerable interest in the work of these aspects due to their relationship with physical and mental health, highlighting the fact that emotions can influence them [25–29].

As has been indicated, these states that are rooted in the person are conceived from childhood, and therefore, it is of great interest to see if, once established, they can be modified over the years through intervention proposals. In this sense, there are several studies that have carried out intervention programs with university students looking for an improvement in aspects such as motivation, reduction of anxiety levels, stress or well-being [30–38]. Additionally, Education for Sustainable Development (ESD) proposes to develop competencies that empower people to reflect on their own actions and thus act in complex situations in a sustainable way, although this requires venturing into challenges and actions. It is a holistic and transformative education that creates interactive and student-centered teaching and learning contexts [39]. In future Physical Education professionals, if these aspects of personal development, wisdom and emotional well-being are worked on, all the key transversal competencies can be developed to achieve all the Sustainable Development Goals (SDG), such as systemic thinking and anticipation; normative, strategic and collaborative skills; as well as critical thinking, self-awareness and problem solving. In this way, they will be able to transfer these skills to their students, since this subject aims at transversality through tasks that are fun and therefore help children to acquire knowledge in a simple way, including behaviors that promote health in their daily routines, deciding and acting in favor of the promotion of this and the well-being of all.

In the study plans for the degree of specialist teacher in Physical Education, specific competencies focused on the training of wisdom and personal development are not contemplated. There is a gap in this regard, and one of the necessary changes in the educational system is to relativize the importance of data, information and knowledge in general and give much greater weight to wisdom. We know without a doubt that we are in a time of great change and that this rate of change must continue towards education for sustainable

development. Therefore, we need to provide students with tools that allow them to adapt to change and be able to take advantage of the positive opportunities that change can offer them, understanding that wisdom can be applied in health education and in the learning process itself throughout life.

Thus, our research question is: Could the wisdom and emotional well-being of students of the title of specialist teacher in Physical Education be improved through an intervention to promote personal development? Therefore, we hypothesize that an intervention based on proposals of personal knowledge towards oneself and towards our actions with others improves wisdom oriented towards personal development and awareness, as a commitment towards an education for sustainable development, in students of the title of specialist teacher in Physical Education.

ESD seeks a transformative and action-oriented pedagogy. Thus, the main objective of this study is to verify if an intervention based on proposals of personal knowledge towards oneself and towards our actions with others improves the wisdom oriented towards personal development and awareness as an undertaking towards an education of sustainable development in students of the qualification of specialist teacher in Physical Education. On the other hand, the second objective focuses on evaluating how this intervention can influence the three areas of wisdom, known as cognitive, affective and reflective areas and, in turn, check if there are differences after the intervention based on the gender of the participants. All this is in the aim of improving their psychological health and emotional well-being and transmitting this way of feeling and acting to primary and secondary students in their future work.

## 2. Materials and Methods

### 2.1. Study Design and Participants

This is a quasi-experimental study with pre- and post-test measurements. These measurements were made in third- and fourth-year students of the degree of specialist teacher in physical education who are taking the subject of "Body expression and communication" at the University of Jaen. A total of 100 students completed the pre-test and the post-test and participated in the present study, 60 being women and 40 being men. This study had the approval of Ethical Committee of University of Jaen (DIC.20/5.TES), and all students voluntarily participated in the study.

### 2.2. Instruments

A multiple response questionnaire was used, where the first question was asked about the course they were in, their date of birth and the gender of the participants. Next, the "Three-Dimensional Wisdom Scale" was used, with the main purpose of measuring the wisdom of the participants.

Three-Dimensional Wisdom Scale (3D-WS) [40]: A scale that seeks to measure 3 dimensions in relation to wisdom, based on the Ardelt scale in 2003 and validated and translated into Spanish by García-Campayo [40]. The scale is made up of three subdimensions with a total of 39 items: 14 items for the cognitive dimension (e.g., "I am hesitant about making important decisions after thinking about them"), 12 items for the reflective dimension (e.g., "When I look back on what has happened to me, I can't help feeling resentful"), and 13 items for the affective dimension (e.g., "I don't like to get involved in listening to another person's troubles"). The items are self-rated using 5 options, and they are scaled from 1 (strongly agree or definitely true of myself) to 5 (strongly disagree or not true of myself). The reliability values for this study were $\alpha = 0.816$ and $\alpha = 0.846$ for the pre-test and post-test, respectively.

With regard to distributing the questionnaires, they were administered electronically together with the instructions to a total of 106 students (finally, 100 took the pre-test and the post-test) in a quiet environment and were answered in approximately 15 min. They were urged to be honest as they would be anonymized afterwards and answers would not influence their academic grades.

### 2.3. Procedure and Intervention

In the first place, the intervention proposal was prepared by seeking an improvement in personal development and awareness, with a total of 13 weeks, including 8 interventions of an hour and a half in duration in subject expression and body communication. The test was administered both initially and after the intervention to verify the results. More specifically, the development was as follows:

The first week, the participants were informed about the procedure to follow to complete the questionnaires in the initial evaluation, and this procedure was carried out in the same way at the end of the process. They were offered the instructions to access it by telematic means, and an operational delivery time was established with a maximum of 15 min. In the eight intermediate weeks, the methodological proposal was developed where sessions were held, oriented to the points of improvement that had been marked as the objective of the study. The students had previously had a document that provided information of interest associated with the activities to be carried out in said practice, which were studied and then responded to individually. Next, the subject joined a working group where the information of all the members of that group was put in common, and common guidelines and patterns were established by making a brief report that had a scheduled delivery. More specifically, we can see this in Table 1.

**Table 1.** Sessions' chronogram.

| Weeks | Activity Explanations |
| --- | --- |
| 1st (1 February 2021) | Explanation. We analyze what determines our responses |
| 2nd (8 February 2021) | Initial evaluation. "3D-SW" |
| 3rd (15 February 2021) | Tools for self-control (parts 1, 2 and 3) |
| 4th (22 February 2021) | Tools to avoid what hurts us |
| 5th (1 March 2021) | Day off Autonomous work |
| 6th (8 March 2021) | Tools to create improvement habits |
| 7th (15 March 2021) | Tools to develop silence |
| 8th (22 March 2021) | Tools to improve relationships |
| 9th (29 March 2021) | Day off Autonomous work |
| 10th (5 April 2021) | The Universal Laws applied |
| 11th (12 April 2021) | Tools to interpret internal processes |
| 12th (19 April 2021) | Control and monitoring instruments |
| 13th (26 April 2021) | Final evaluation. "3D-SW" |

Taking into consideration guidelines from various authors [41–45], the methodological process follows a sequence or structure that seeks to reduce, until their final elimination, those influences that harm the person, such as training the mind to stop living in the past or in the future, stop judging ourselves and others, letting go of control of what is not controllable and allow life to rule, learning to relativize what happens under the norms of human justice, seeking the elimination of dependencies towards people and means with sensationalist speeches that are not accompanied by positive final intention and do not convey solutions and seeking to prevent emotional moments and thoughts that may affect what is of great value in your life.

Once aware of the procedures for the elimination of these patterns, we aim to carry out, on a daily basis, those things that help us to improve and make proposals that help us accept what happens using universal laws to guide us [2], seeking solutions to the situations we experience in our destiny and with a life purpose and through which a scale of priorities is established that allows us to discern aspects of events, forgiving ourselves, accepting failures and weaknesses and trusting life. To improve functionality and consolidation, protocols of mental silence or contemplation are shown to help focus attention [46,47]. Verbalizing the positive despite feeling artificial at the beginning and using gratitude in everything is part of the development of this stage.

Finally, in a personalized way and to favor the discovery of an objective reality, personalized and individual supports are proposed, providing tools for self-knowledge.

The tools are focused on improving mental silence, identifying purpose, recognizing universal laws, reducing what hurts us, creating improvement habits and developing empathy in social relationships.

As an example, following the book by Arteaga [48], some of the proposals that were carried out are specified in Table 2.

**Table 2.** Exercises examples and description.

| Exercise | Description and Purpose |
| --- | --- |
| "Enneagram" | It is a system to classify the personality and includes 9 types or ways of seeing the world called enneatypes. It offers a roadmap for analyzing personality patterns and helps to identify them. Two groups are divided, and in each group, they must choose the paragraph that best describes their attitudes and behavior. Finally, they are classified into an enneatype (enthusiastic, challenger, peacemaker, etc.). |
| "A happy world" | This activity consists of devising a world around us where everything that happens is good and perfect. Visualize yourself in it and discover an image of how you would be. Finally, one must describe the image that has been seen. |
| "The crude reality" | Students are instructed to perform the following: visualize real life and list the factors that prevent you from being the image you saw in the "brave new world" exercise. Are they external factors to you? |
| "The system" | The following is indicated: Consider your body as a working system or machine, or as if it were a company working together on a global project. They must visualize aspects such as: What would be the best and most appropriate action to be effective when there are people who do not perform their function? Imagine parts of your body that do not do their function. |
| "The wheel of life" | Draw a circle and divide it into six equal segments, leaving a circle with six sections that represent the six main areas of life. There could be more, but those six vital areas are universal. These areas can be evaluated and become six goals for an ideal life (self-knowledge, health andwell-being, emotions, relationships, transcendence and professional). Each area is scored from 1 to 10. |
| "Reality on my Screen" | Students are asked the following: Write about yourself, the things you give yourself and give yourself permission aimed at creating an image of yourself that is real and in harmony with what you want to be. What do you want to experience more, what gives you confidence and energy, who would you like to be? |

The sequence proposed to promote development and personal identity would consist, first of all, in creating and reflecting on everyday situations in which suffering and discomfort appear and are felt, producing mistakes and errors in the actions that we carry out. Then, one learns to recognize the existence of natural laws and their operation to realize and perform our own actions and have a reference pattern. At that point, one becomes aware of the individual unconscious defense patterns, since it is useful to recognize the motivations that arise from the mechanisms of the ego, to provide different perspectives of reality and how to face it and, lastly, to provide guiding references.

### 2.4. Data Analysis

First, we checked the internal consistency of both the pre-test and the post-test of each of the scales, using Cronbach's alpha test to calculate their reliability. Then, we checked the normality of the variables using Kolmogorov–Smirnov and chi-square tests.

After that, a T-Student test for related measures was conducted. Additionally, we checked the differences between gender using a MANOVA (Multivariate Analysis of Variance). An analysis of the residuals revealed the non-fulfillment of the hypothesis of normality and homoscedasticity of some variables, so it was decided to carry out the analyses also using non-parametric tests (Wilcoxon test and Mann–Whitney U test). The results obtained with both procedures were very similar; therefore, the results of non-parametric tests were not included for brevity. The statistical package IBM SPSS 24.0 (New York, NY, USA) was used for the analysis. The statistical values used were F to MANOVA (multivariate effect of the relation analyzing if the variability between the means of the groups is greater than the variability of the observations within the groups. Table 3) and $p$ for MANOVA and t-Student tests (Table 2 and Table 3). A value of $p < 0.05$ is statistically significant. It indicates strong evidence against the null hypothesis, as there is less than a 5% probability that the null is correct (and the results are random). Therefore, we reject the null hypothesis and accept the alternative hypothesis [49].

## 3. Results

In the first place, taking into account the total sample analyzed, we find the results reflected in Table 3.

**Table 3.** Analysis of pre- and post-tests for the sample.

| | 3D-Wise Scale | | |
| --- | --- | --- | --- |
| | **M** | **SD** | *p* |
| Ambit_Affective_Pre | 3.53 | 0.49 | |
| Ambit_Affective_Post | 3.49 | 0.48 | 0.414 |
| Ambit_Reflective_Pre | 3.47 | 0.50 | |
| Ambit_Reflective_Post | 3.58 | 0.51 | 0.038 * |
| Ambit_Cognitive_Pre | 3.43 | 0.48 | |
| Ambit_Cognitive_Post | 3.44 | 0.55 | 0.754 |
| Ambit_Global_Pre | 3.47 | 0.38 | |
| Ambit_Global_Post | 3.50 | 0.41 | 0.489 |

Note: * $p < 0.05$; M = Mean; SD = Standard Deviation.

Considering the results, we found no changes after the intervention weeks in the affective or cognitive variables, where the values remained practically unchanged. On the other hand, in the reflective field, we did find statistically significant differences ($p = 0.038$) for the total sample.

The multivariate analysis carried out to differentiate according to gender reflected the results indicated in Table 4. This analysis was carried out, on the one hand, for the three variables of the questionnaire in total and, on the other hand, for the scale of the questionnaire as a whole, taking into account the results of the Univariate ANOVA.

**Table 4.** Multivariate analysis of the variables according to gender.

| | **Men** | | | **Women** | | | | |
| --- | --- | --- | --- | --- | --- | --- | --- | --- |
| | **M** | **SD** | *p* | **M** | **SD** | *p* | **F** | *p* |
| Ambit_Affective_Pre | 3.68 | 0.43 | | 3.43 | 0.51 | | 5.915 | 0.017 * |
| Ambit_Affective_Post | 3.63 | 0.51 | 0.670 | 3.4 | 0.44 | 0.584 | 5.670 | 0.019 * |
| Ambit_Reflective_Pre | 3.54 | 0.35 | | 3.43 | 0.58 | | 1.134 | 0.290 |
| Ambit_Reflective_Post | 3.66 | 0.47 | 0.127 | 3.52 | 0.52 | 0.237 | 1.310 | 0.255 |
| Ambit_Cognitive_Pre | 3.48 | 0.51 | | 3.4 | 0.46 | | 0.736 | 0.393 |
| Ambit_Cognitive_Post | 3.57 | 0.54 | 0.140 | 3.36 | 0.54 | 0.549 | 3.467 | 0.066 |
| Ambit_Global_Pre | 3.56 | 0.35 | | 3.42 | 0.40 | | 3.385 | 0.034 * |
| Ambit_Global_Post | 3.62 | 0.43 | 0.328 | 3.43 | 0.40 | 0.853 | 4.861 | 0.048 * |

Note: * $p < 0.05$; M = Mean; SD = Standard Deviation.

Regarding gender, we found statistically significant differences both in the pre-test and in the post-test for the affective domain variable, where men had statistically significant higher values than women ($p$ = 0.017 and 0.019). In addition, in the post-test cognitive field, they had an approximation to significance ($p$ = 0.066). Furthermore, the scale as a whole showed statistically significant differences in favor of men at $p$ = 0.034 and 0.048 (pre-test and post-test). Finally, to check if the intervention had a different influence on boys and girls, we proceeded to segment the file and make the comparison between the pre-test and post-test separately according to gender. In this way, it was possible to see how no statistically significant differences were found, which could be due to the small sample (40 and 60 subjects), if observing how men had a similar value to that of the reflective field (which, statistically, gave differences when joining the sample) in the cognitive field.

## 4. Discussion

The objective of this study was firstly to verify if an intervention based on proposals of personal knowledge towards oneself and towards our actions with others improves the wisdom oriented towards personal development and awareness as an undertaking towards an education of sustainable development in students of the qualification of specialist teacher in Physical Education. The second objective focuses on evaluating how this intervention can influence the three areas of wisdom, known as cognitive, affective and reflective areas, and, in turn, check if there are differences after the intervention based on the gender of the participants.

In university students, it should be noted that this is the first study using this scale, but the interventions carried out in order to improve certain related variables show quite disparate results and make us reflect on the importance of having an adequate protocol to intervene. So, we find investigations that did not achieve improvements in satisfaction and learning [50], and other studies did find improvements in academic satisfaction [31,51] or with life satisfaction [37] with similar intervention durations. This aspect was not assessed in the present study, but it is a very interesting aspect to take into account.

Entering psychological variables, interventions in university students have given good results in aspects such as reducing levels of anxiety, stress and depression [33]. It would be very interesting to consider the hybridization of methodologies that provide an optimal result due to their multiple possibilities [52], for example, combining the current intervention with biofeedback, which has given very good results in university students in variables related to emotional well-being in interventions from as little as 4 weeks [33,34,53] and with other aspects as important to life as resilience or the ability to overcome adversity [38].

Describing the students of the degree in Education linked to Physical Education, the promotion of healthy habits related to physical activity is essential. In this regard, in recent years, an increase has been observed due to the study of those factors associated with a healthy lifestyle of the young population, especially university students [54]. Therefore, interventions where this stimulation is sought, such as that of Carranza [36], with more than 800 students, may be appropriate to promote these aspects and teach university students who aspire to be future teachers.

Focusing on the first of the objectives, the intervention strategy used by the study has aimed at creating a transformation or change in the areas of "Self-knowledge", "Emotions", "Relationships" and "Transcendence of life", avoiding the "learned helplessness" that consists of looking for something where we cannot really find it or, what is the same, looking for it outside when the answer is within us. This concept conveys the need to change the paradigm we have for understanding life, which is usually associated with seeking solutions, answers and changes outside of ourselves.

Montávez [16] establishes that this awareness is the first phase in the personal development process and consists of "perceiving and feeling the body and becoming aware of its reality", and adds that awareness is carried out through their own personal experiences, that is, through "experimentation". This phase is developed in our intervention constantly

since each experience lived is taken into account and would end when the individual becomes capable of adapting the work carried out to their personal characteristics, which in this research, would be when arriving at the last session.

The students' perception of the intervention in general was positive after carrying out surveys during the classes about the usefulness they saw in the exercises and in the way of carrying out the sessions compared to the rest of the academic course in relation to the subject. Therefore, it is thought that it may be an appropriate protocol for students, which must be deepened so that they can also apply it safely to their students in the future. Secondly, when describing the results obtained after the intervention, we must emphasize that statistically significant changes were observed in the reflective field, taking into account the total sample.

The second purpose of this study was in relation to assessing how this intervention could influence to a greater or lesser extent in the areas of wisdom and whether there could be differences according to gender. Although the measurement of wisdom through questionnaires may be simplistic because it is thought that it is affected by biases such as social desirability [50], it does have advantages, offering a simple and fast means to approach the base empirically and allowing us to find out antecedents, starting situations and consequences. It should be noted that there are different investigations that have translated the 3D-WS scale into different languages, but they are very recent studies [40,55,56].

Our intervention greatly promoted the participants' self-knowledge and reflection on the events and experiences they had had, and it seems logical that this was the greatest change experienced by them. In this sense, it is important to explain that this program focuses substantially on self-knowledge and reflection, so this could be an explanation. No studies have been found that have analyzed this area in university students with this questionnaire, but correlations have been identified between the global score of this questionnaire with happiness [57], life satisfaction [58] or emotional competence and self-efficacy [59]. This is very interesting to consider given that the observation of low levels of these scales could be detrimental to these aspects.

With regard to gender, we observed that men had higher values than women both in the pre-test and in the post-test, although there were no differences in the change (both improved, but they were not statistically significant data). It could be of great interest to investigate why these values are higher in men and to make intervention proposals especially focused on the female gender. Furthermore, this differs from Andreia's study [60] with this scale where he indicates the non-existence of differences based on gender or educational level, generally finding that young people score higher in the three dimensions than older people [61]. However, it should be noted the study by Fernandez [62], where differences according to age are observed in the affective and reflective dimensions but not in the cognitive one, reflects similar results to our study.

We must point out that there is an urgent need to accelerate actions to achieve the sustainable development goals that are within the 2030 Agenda and, without the involvement of universities, these cannot be met [63]. Among them are guaranteeing a healthy life and promoting well-being at all ages, as well as guaranteeing an inclusive, equitable and quality education. Future teachers can provide solutions, knowledge and innovative ideas to the sustainable development goals, mainly those focused on healthy habits, training those who will be in charge and responsible for implementing these goals [62]. They need to innovate by mobilizing, involving and motivating the youngest; these, as in the university, give a quality education with relevant skills that promote and research on sustainable development. This will lead to better personal, national and international development results.

There are difficulties in finding research or studies that relate the influence that the teaching of personal development and self-awareness exerts on wisdom in its cognitive, reflective and affective spheres, with there being very few authors who have carried out studies on it, and there are none at the university level. The intervention presented in this study can lead to an improvement in the process of personal development and awareness

of the participants thanks to the program of self-awareness. It can be useful for creating habits of improvement towards ourselves and with our social relationships. As the UN Secretary General points out in relation to working with the SDGs, "if we want to improve, we must act differently." It is possible that many of the objectives will be achieved, or at least advanced by them, if there are real changes in attitudes and behavior [64].

On the other hand, from the information compiled in the bibliographic review, the conclusion is reached of the importance of addressing personal and conscientious development in the educational field as a channel to externalize one's own feelings and emotions. Likewise, personal knowledge acquires great relevance for this process, becoming a primary phase. The object of another study would be to verify the different educational systems established in Spain and see the relationship it has with the new trends based on Gardner's Multiple Intelligences, where developing emotional competence has become a significant aspect. We must help students to recognize themselves as human persons and act as such, from the field of health in this case. In this task, education is one of the most important pillars and the basis for the realization of the 2030 Agenda [65]. We must also foster in them a good teacher–student relationship based on mutual respect, justice and trust [66] and promoting good physical, mental and social health for all.

*Limitations*

A series of limitations should be noted following the Engberg recommendations [67], such as the need to randomize the participants, carry out mixed-method studies including a qualitative analysis or increase the duration of the intervention. Moreover, we did not have a control group, and the sample size was low, and this could influence the significance of results. Finally, there were no changes in two of the areas under study. In this sense, it is suggested that more social work proposals be included, for instance, with the development of laughter therapy sessions or to getting to know colleagues to improve affective ambit and executive function work with gamification in classes in the cognitive ambit.

## 5. Conclusions

The intervention that was carried out could serve to improve emotional well-being and wisdom processes in university students. There was a positive but not significant trend in most of the studied aspects of wisdom, improving the reflective field, although more studies are necessary in this field. In addition, women had lower results than men, which makes reflection on this aspect also important.

It is necessary to emphasize the importance of continuing to carry out studies and research on related aspects, since there are clear benefits that can be produced for. More studies are also recommended, especially intervention studies, to test improvements in the general wisdom of the university environment, where longitudinal studies of larger and more diverse samples are advisable to explore the mediating effects of these constructs on well-being, respect and learning.

**Author Contributions:** Conceptualization, M.A.-C. and M.V.P.-M.; methodology, M.A.-C.; formal analysis D.M.-S.; investigation M.A.-C. and D.M.-S.; data curation D.M.-S.; writing-original draft preparation M.A.-C. and D.M.-S.; writing—review and editing, M.A.-C., M.V.P.-M. and D.M.-S. All authors have read and agreed to the published version of the manuscript.

**Funding:** This research received no external funding.

**Institutional Review Board Statement:** The study was conducted in accordance with the Declaration of Helsinki, and approved by the Institutional Review Board (or Ethics Committee) of University of Jaen (DIC.20/5.TES).

**Informed Consent Statement:** Informed consent was obtained from all subjects involved in the study.

**Data Availability Statement:** Not applicable.

**Conflicts of Interest:** The authors declare no conflict of interest.

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
