# Peer review of "Application of a Program to Improve Personal Development in Future Physical Education Teachers of the Degree in Education and Its Relationship with Wisdom"

_sustainability, doi:10.3390/su14031188_

Round 1

Reviewer 1 Report

Dear all

Thank you for the opportunity. I wrote my review via Word and uploaded it to attachments.

Author Response

We attach the answer to the reviewer. 

Thanks you for your appreciations.

Reviewer 2 Report

The topic of the article is very interesting. My recommendations are:
In the abstract, mention descriptions of 3S-WS.
I recommend that in the introduction you motivate why you chose your students from the Physical Education specialization.
In tables 2 and 3 I recommend rearranging the value of p, between the two rows of the evaluation. This is how it is calculated that you only calculated p on the pretext.

Author Response

Dear reviewer, thanks you for your comments. The authors are very grateful for your suggestions in order to improve the manuscript.

- We have clarifies 3S-WS scale in introduction

- We have included why we select PE students (lines 116-127)

- We are changed the row of p-test (from pre-test to post-test). On the other hand, we have included some reference to the adequate value of p (reference 48)

Reviewer 3 Report

Dear author(s),                                                              

I have read with much interest the manuscript titled “Application of a program to promote personal development and self-awareness in future Physical Education teachers of the degree in Education and its relationship with wisdom”. The article presents some data regarding the influence of a program, which promotes personal development and self-awareness, on wisdom among future Physical Education teachers.

However, there are some issues that I noticed must be clarified in order to I be able to recommend the paper for publication.

Please find below major points in the communication which needs clarification / reanalysis, rewrites and/or additional information and suggestions for what could be done to improve the paper.

GENERAL RECOMMENDATIONS

I strongly suggest:

  1. a better and clearer highlight the connection of this study with the health and sustainability (see also the purpose of the journal, respectively "Health and sustainability" section, and the “Innovations and Trends on Higher Education and Sustainable Development in the Field of Health Sciences” special issue purpose).
  2. clarification (and alignment) of the purpose of the study (see lines 10-12, 112-118, 243-246)
  3. to clearer highlight the research question(s) which the purpose of the research is addressed.
  4. to clearer highlight whether the present research complements certain deficiencies/gaps existing in recent research in the field, OR resolves certain confusions and which are these confusions/deficiencies/gaps (A more consistent literature study would be needed).
  5. to reinforce the results section with bibliographic references regarding the value of p accompanied by additional explanations regarding the establishment of the (value for) existence of statistically significant / insignificant differences (see also the values highlighted in Table 2, 3 and in the highlighted Notes).
  6. to reinforce the discussion section by a more specific interpretation of the results (in relation to research question(s) and in the context of more other evidences).
  7. to discuss in a separate section the limitations of the study
  8. to revise the conclusion section in order to better highlight how the obtained results respond to the addressed problem/to the proposed aim in the Conclusion section (address the proposed aim and clearly summarize the answer to the research questions).
  9. to revise of the English language.

SPECIFIC RECOMMENDATIONS

Line 12: I recommend to detail the “3S-WS” abbreviation

Line 12: I recommend to align the number of weeks (see lines 150,156)

Line 120-125: I recommend to highlight in this section the main stages/steps for overall research design and description of participants in a separate section

Line 227: I recommend to explain the significance of all parameters highlighted in the table (e.g. F, the two last columns)

Author Response

Dear reviewer, thanks you for your comments. The authors are very grateful for your suggestions in order to improve the manuscript, which we have tried to do attend to in the most meticulous way possible.

Below your suggestions we explain the answers and changes:

1- The introduction has been changed including a link between health, sustainability and the ESD (Education for Sustainable Development) purposes. 

2- The purpose at the end of introduction and starting discussion section has been modified and clarified.

3/4- We have included some new references about this topic. But, the novelty of this study is that there is a knowledge gap in the use of this kind of intervention specially in university students. This is why the researchers decided to carry out this study, only in the field of clinical psychology but not intervention of this kind of study. We have included more references about this.

5- The statistical analysis and results sections have been modified and a statistical reference has also been included.

6/8- We have modified this section including the justification for the use of  university students (277-305), the first  purpose (lines 306-327)  and the second goal (328-355) and practical applications with 2030 Agenda and SDGs (353-374). Finally we have included a limitation section at the end of the discussion (389-396)

9- We have reviewed the language with a expert native and all changes have been included in the manuscript

Specific recommendations

Line 12: I recommend to detail the “3S-WS” abbreviation

It has been modified

Line 12: I recommend to align the number of weeks (see lines 150,156)

It has been clarified with table 1 and a description about the intervention weeks.

Line 120-125: I recommend to highlight in this section the main stages/steps for overall research design and description of participants in a separate section

It has been clarified with table 1 and a description about the intervention weeks (2.3 procedure and intervention)

Line 227: I recommend to explain the significance of all parameters highlighted in the table (e.g. F, the two last columns)

It has been included in the table and in the statistical analysis section (reference 48)

Best regards,

Round 2

Reviewer 1 Report

Dear Authors

It seems that the manuscript has been developed very much. It seems very consisted now. However, is still have few concerns and recommendations.

  1. I still do not see clear research questions. Could formulate them? Or alternatively, formulate clear hypotheses. You use the word hypothesis 4 times in the manuscript but none of them is related in describing them They address the approval or rejection.
  2. Figure 1 must be cited.
  3. Regarding to the research gap. I strongly recommend you introduce the aim of this research as the solution on contributing to the found research gap. This is a matter that needs to be clearly introduced in the introduction.

After these, the paper can be accepted.

Author Response

Dear reviewer, 

Thanks you for your appreciations. We have include the hypothesis about this study.

Thus, our research question is: Could the wisdom and emotional well-being of students of the title of specialist teacher in Physical Education be improved through an intervention to promote personal development? (Lines 143-145).

Therefore, we hypothesize that an intervention based on proposals of personal knowledge towards oneself and towards our actions with others improves wisdom oriented towards personal development and awareness, as a commitment towards an education for sustainable development, in students of the title of specialist teacher in Physical Education (Lines 145-149).

On the other hand, figure 1 has been cited.

Finally, according to the 3º aspect, we have include a new description:

In the study plans for the degree of specialist teacher in Physical Education, specific competencies focused on the training of wisdom and personal development are not contemplated. There is a gap in this regard and one of the necessary changes in the educational system is to relativize the importance of data, information and knowledge in general and give much greater weight to wisdom. We know without a doubt that we are in a time of great change and that this rate of change must continue towards education for sustainable development. Therefore, we need to provide students with tools that allow them to adapt to change and be able to take advantage of the positive opportunities that change can offer them, understanding that wisdom can be applied in health education and in the learning process itself throughout life (Lines 133-142).

Thanks you for this opportunity to improve the manuscript.

Reviewer 3 Report

Dear author(s),                                                              

I have read with much interest the revised manuscript (re)titled “Application of a program to improve personal development in future Physical Education teachers of the degree in Education and its relationship with wisdom”. The article has been improved in relation to the recommendations received.

However, I suggest correlating the applications in the text with the values highlighted for p in paragraphs 257-261 and in lines 265, 276 (notes associated with the tables).

Author Response

Thanks you to the reviewer.

We have change this aspects and changed the p value (all are .05 what is right)

Best regards.